# A green solvent enables precursor phase engineering of stable formamidinium lead triiodide perovskite solar cells

Benjamin M. Gallant[1,2], Philippe Holzhey[1], Joel A. Smith[1], Saqlain Choudhary[1], Karim A. Elmestekawy [1], Pietro Caprioglio[1], Igal Levine[3,4], Alexandra A. Sheader[1], Esther Y-H. Hung[1], Fengning Yang[1], Daniel T. W. Toolan [5,6], Rachel C. Kilbride [6], Karl-Augustin Zaininger[1], James M. Ball[1], M. Greyson Christoforo [1], Nakita K. Noel [1], Laura M. Herz [1,7], Dominik J. Kubicki [2] & Henry J. Snaith [1] ✉

Perovskite solar cells (PSCs) offer an efficient, inexpensive alternative to current photovoltaic technologies, with the potential for manufacture via high-throughput coating methods. However, challenges for commercial-scale solution-processing of metal-halide perovskites include the use of harmful solvents, the expense of maintaining controlled atmospheric conditions, and the inherent instabilities of PSCs under operation. Here, we address these challenges by introducing a high volatility, low toxicity, biorenewable solvent system to fabricate a range of 2D perovskites, which we use as highly effective precursor phases for subsequent transformation to α-formamidinium lead triiodide (α-FAPbI$_3$), fully processed under ambient conditions. PSCs utilising our α-FAPbI$_3$ reproducibly show remarkable stability under illumination and elevated temperature (ISOS-L-2) and "damp heat" (ISOS-D-3) stressing, surpassing other state-of-the-art perovskite compositions. We determine that this enhancement is a consequence of the 2D precursor phase crystallisation route, which simultaneously avoids retention of residual low-volatility solvents (such as DMF and DMSO) and reduces the rate of degradation of FA$^+$ in the material. Our findings highlight both the critical role of the initial crystallisation process in determining the operational stability of perovskite materials, and that neat FA$^+$-based perovskites can be competitively stable despite the inherent metastability of the α-phase.

Scalable, reproducible and controlled crystallisation of highly stable metal halide perovskite thin films is crucial to the commercial implementation of this promising class of semiconductor materials in optoelectronic applications. Ideally, solution-processed perovskite deposition should employ inks based on non-toxic and volatile solvents and should result in the controlled formation of a well-mixed and low defect perovskite phase that – crucially – possesses long term operational stability under external stimuli such as heat, light and moisture[1–3].

[1]Clarendon Laboratory, Department of Physics, University of Oxford, Parks Road, Oxford OX1 3PU, United Kingdom. [2]School of Chemistry, University of Birmingham, B15 2TT Birmingham, UK. [3]Solar Energy Division, Helmholtz-Zentrum Berlin für Materialien und Energie GmbH, Berlin 12489, Germany. [4]Institute of Chemistry and The Center for Nanoscience and Nanotechnology, The Hebrew University, Jerusalem 91904, Israel. [5]Department of Materials, University of Manchester, Manchester M13 9PL, UK. [6]Department of Chemistry, University of Sheffield, Sheffield S3 7HF, UK. [7]Institute for Advanced Study, TU Munich, Lichtenbergstr. 2a, 85748 Garching, Germany. ✉e-mail: henry.snaith@physics.ox.ac.uk

Significant attention has been paid to the relationship between halide perovskite composition (ABX₃ – where A is an organic or alkali-metal cation, B is a lead or tin cation, and X are halide anions) and operational stability. Exchanging the archetypical A-site methylammonium (MA⁺) cation for formamidinium (FA⁺) significantly increases thermal and photochemical stability but introduces a phase instability. The photoactive α-FAPbI₃ phase is only metastable at room temperature, and the transition to the yellow non-perovskite δ-phase is accelerated by the presence of moisture[4]. This instability has been circumvented by alloying FA⁺ with caesium cations (Cs⁺)[2]. However, A-site inhomogeneity has been linked to both poor optoelectronic performance[5] and other long-term degradation pathways, such as the emergence of Cs⁺-rich impurity phases[6,7]. Further, of all the elements employed in halide perovskites, the scarcity of Cs poses the greatest challenge in availability for TW scale PV production[8] and thus routes realising stable neat-FA⁺ perovskites are greatly advantageous.

However, despite the emphasis on ABX₃ composition as the key factor controlling perovskite operational stability, properties such as the types[9] and density[10] of defects, impurity phases[11], compositional inhomogeneity[6], residual solvents[12–14] and residual strain[15] have also been shown to influence long-term stability. All such properties are imparted into the perovskite thin layer during crystallisation of the ABX₃ structure, which can proceed via a range of intermediates or precursor phases, including solvate[16–18] and polytype[7,19] phases and phases incorporating sacrificial ions[20]. Given the origin of these properties, it is imperative that the relationship between crystallisation pathway, precursor phase formation and resulting ABX₃ perovskite stability be investigated and understood more widely.

Despite substantial research over the past decade[21,22], highly toxic N,N-dimethylformamide (DMF) remains ubiquitous in halide perovskite inks, while operational instability of perovskites-based optoelectronics remains a key barrier to their wider application[1]. Here we simultaneously combat the challenges of perovskite ink toxicity, processing-induced instability and compositional heterogeneity. By developing a precursor ink that utilises a highly volatile, low toxicity solvent mixture, we replace conventional precursor phases with a solid-state 2D perovskite material. By exchanging the organic cations in the 2D precursor phase with FA⁺, we enable the subsequent

crystallisation of α-FAPbI₃ perovskite. We not only find that our α-FAPbI₃ performs effectively in PSCs, but that 2D-precursor phase growth produces α-FAPbI₃ with substantially improved ambient, thermal and photostability in comparison to any other perovskite composition or processing route that we fabricate herein[2,3]. We are able to fabricate neat FAPbI₃ thin films that are stable for >3000 hours under harsh heat, light and moisture conditions, and achieve a promising champion lifetime to 80% of initial performance (t₈₀) of 800 hours and no degradation (t₁₀₀) for more than 1930 hours under ISOS-L-2 (85 °C, 1-sun equivalent) and ISOS-D-3 (85 °C, 85% relative humidity), respectively, when integrated into PSCs. Our work enables us to distinguish how different processing variables impact perovskite stability and demonstrate why conventional solution processing routes are not only problematic for toxicity but are fundamentally linked to halide perovskite instability.

## Results and Discussion
### A biorenewable and low toxicity solvent system
As solution processing at industrial scale dictates that the majority of solvents employed must be volatilised into the atmosphere[23], the hazard of employing toxic solvents in PSC production at increasingly large scale are substantial. In light of this, several DMF-free solvent systems have been established for perovskite solution processing, notably based on N,N-dimethylacetamide (DMAc)[24], γ-butyrolactone (GBL)[25], 2-methoxyethanol (2-ME)[26] and acetonitrile (ACN) in combination with methylamine gas[27]. However, as Fig. 1a shows, DMAc, 2-ME and ACN do not provide the desired reduction in toxicity, while GBL is an illegal narcotic on account of its metabolite, γ-hydroxybutyrate (GHB)[28]. Furthermore, the use of MA gas as a co-solvent, poses further problems of volatility, where escape of the MA gas can render the solution unstable and pose a toxicology risk itself. Moreover, while dimethyl sulfoxide (DMSO) – the typical companion to DMF in perovskite solution processing – is non-toxic as a neat liquid, in the context of lead salt solvation the proficiency of DMSO as a percutaneous absorption (skin-penetration) enhancer[29,30] is of acute concern for at-scale processing (Fig. 1b).

Here we investigate and develop a solvent system consisting of biorenewable cyclic ethers and alkylamine liquids, in which the latter serves both as a coordinating co-solvent and subsequently as a component within the 2D perovskite precursor phase. This system can be considered part of a family of solvent mixtures all comprising poorly Pb²⁺-coordinating solvents as the majority component alongside strongly Pb²⁺-coordinating solvents[26]. Here cyclic ethers take on the role of the poorly coordinating solvent while butylamine (BA), which is liquid at room temperature, acts as the strongly coordinating solvent. We find that both tetrahydrofuran (THF) and 2-methyltetrahydrofuran (MeTHF) are suitable poorly-coordinating solvents, however we focus on the latter on account of its biorenewable production, significantly lower toxicity and favourable vapour pressure for rapid at-scale solution processing. Vidal, et al[23]., recently highlighted that THF's high vapour pressure renders its lifetime environmental impact relatively large compared with less volatile perovskite solvents due to additional greenhouse gas emissions resulting from energy-intensive solvent recapture. However, a lifecycle analysis of MeTHF conducted by Slater, et al[31]., demonstrated a 97 % reduction in lifetime emissions associated with the use of MeTHF as compared to THF, due mostly to its biorenewable production and significantly lower vapour pressure. Moreover, MeTHF has recently been recommended by both the European Medicines Agency (EMA) and the Food and Drug Administration (FDA) as a Class 3 solvent ("solvents with low toxic potential") with a permitted daily exposure limit of 50 mg day⁻¹ [28]. This exposure limit is much higher than that of THF (7.2 mg day⁻¹), which Vidal, et al[23]., demonstrate already possesses significant toxicological advantages over other common perovskite solvents. Furthermore, as solution processing demands that the majority of solvents employed are

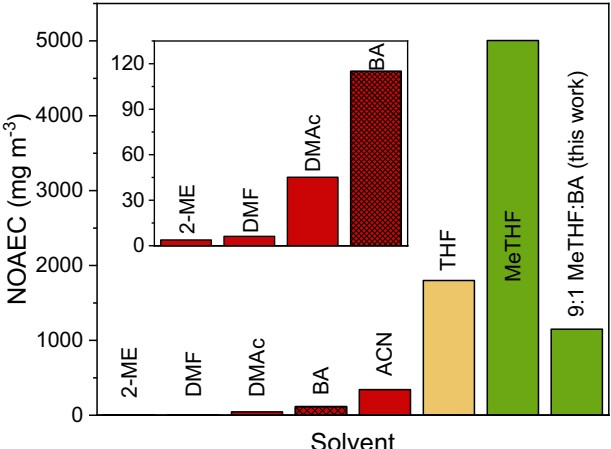

**Fig. 1 | Scalable processing compatibility.** Preferential properties of 2-methyltetrahydrofuran (MeTHF) and THF as solvents in industrial processes where vapour phase exposure is unavoidable. Solvents shown in green are produced industrially from biorenewable sources, in yellow may be produced biorenewably but typically are not, and in red cannot be produced from renewable sources. No observed adverse effect concentrations (NOAECs) are taken from the European Chemicals Agency (ECHA) database and are calculated for workplace (except n-butylamine, BA, where the value for general population is used) exposure via inhalation of solvent vapours.

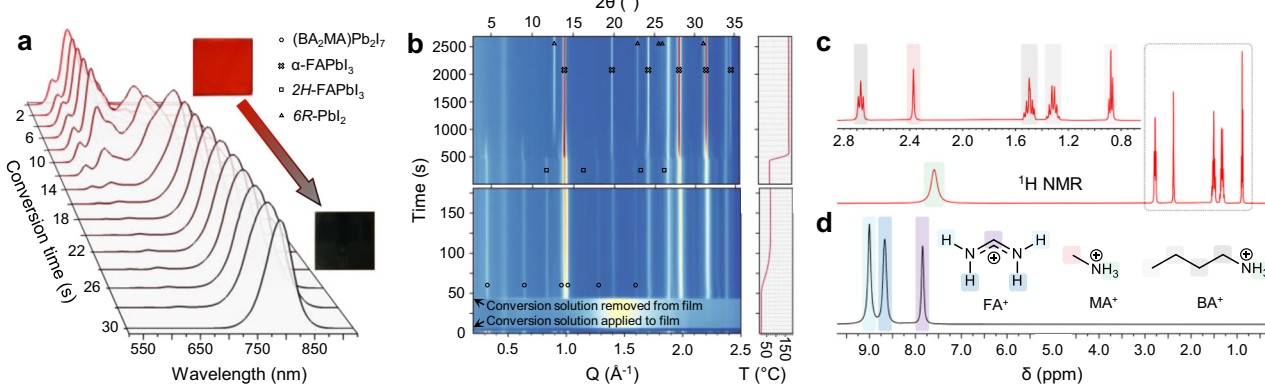

**Fig. 2 | Conversion of 2D precursor phase to 3D α-FAPbI₃ perovskite. a** Tracking photoluminescence (normalised) of optimised 2D precursor phase ($R_{BA:MA+}$ = 1.5) converting to 3D α-FAPbI₃ during soaking in "conversion solution" (FAI solution in *n*-butanol). Unconverted 2D precursor phase shown at t = 0 s. Insets: images of 2D precursor phase (red) and 3D (black) perovskite thin films. **b** In-situ grazing-incidence wide-angle X-ray scattering (GIWAXS) tracking evolution of crystalline

phases present as 2D perovskite is converted to α-FAPbI₃. ¹H solution nuclear magnetic resonance (NMR) spectra of (**c**) 2D precursor phase ($R_{BA:MA+}$ = 1.5) thin films (inset shows expanded region of spectrum corresponding to methyl and methylene signals, highlighted in dashed box) and (**d**) α-FAPbI₃ thin films dissolved into DMSO-$d_6$.

volatilised, the other safety-critical parameters for at-scale processing are the no observed adverse effect concentration (NOAEC) via inhalation and the lower flammability limit, rather than more conventional measures of toxicity, such as the lethal dose (LD50) or concentration (LC50). Based on the European Chemicals Agency's (ECHA) NOAECs (exposure via inhalation) for MeTHF and BA we calculate a NOAEC for our mixed solvent system (9:1 MeTHF:BA) of 1150 mg m⁻³, a 185-fold increase in comparison to DMF (6.2 mg m⁻³) (Fig. 1a). The lower flammability limit of all solvents commonly used for perovskite processing are comparable. Full discussion of these parameters is offered in Supplementary Note 2. Besides these, the critical advantage of both THF and MeTHF over the previously reported ACN systems is that the use of highly volatile MA gas is precluded. Use of a liquid – rather than gas phase – amine minimises the complexity of precursor preparation and handling, improving scalability and ensuring reproducible solution composition.

## 2D precursor phase-growth of 3D perovskites

Notwithstanding the volatility of MA, amines hold their own critical advantage against other Lewis basic Pb²⁺-coordinating solvents; they also behave as Brønsted-Lowry bases in precursor solutions. In our precursor ink the inclusion of BA alongside methylammonium iodide (MAI) leads to a Brønsted-Lowry H⁺ exchange in solution and the in-situ production of *n*-butylammonium (BA⁺). We dissolve MAI and lead (II) iodide (PbI₂) in our MeTHF:BA solvent system and cast a thin film by spin-coating and drying at 70 °C for ten minutes. Incorporation of both BA⁺ and methylammonium (MA⁺) within the perovskite structure leads to highly crystalline 2D Ruddlesden-Popper phase perovskites (RPPs) of the $(BA^+)_2(MA^+)_{n-1}Pb_nI_{3n+1}$ family, where BA⁺ serves as the large A-site cation (A′) and MA⁺ as the small A-site cation (A)[32]. The *n* described by this empirical formula is often used to differentiate RPP phases by the number (n) of lead-halide octahedra ($[PbI_6]^{4+}$) layers interspersed between layers of the large A′ cations. In steady-state photoluminescence (PL), X-ray diffraction (XRD) and absorbance measurements (Supplementary Figs. 2–4) we observe multiple peaks consistent with the presence of multiple RPP phases. By varying BA:MAPbI₃ precursor stoichiometry ($R_{BA:MA+}$) we are able to control the average RPP phase composition, denoted <*n*> (Supplementary Note 3).

Although 2D RPPs have been utilised as photo-absorbers in efficient PSCs, these typically display non-ideal bandgaps for use in single-junction photovoltaics[33]. Instead, we propose an alternative application of 2D mixed-phase RPPs; as solid-state precursor phases, or intermediates, in a solution-processed sequential deposition of 3D

ABX₃ perovskites[34,35]. This strategy allows fabrication of α-FAPbI₃ by our solvent system but avoids the reported reactivity of amines with FA⁺ in solution[36,37].

To convert the 2D precursor phase into α-FAPbI₃, we dispense a solution of formamidinium iodide (FAI) in *n*-butanol ("conversion solution") on top of the substrate and allow conversion via cation exchange to occur over a controlled time before spin-coating. In Fig. 2a we present the PL spectra of the precursor RPP layer as a function of conversion time. Supplementary Fig. 8 tracks the corresponding evolution of absorbance during conversion. While FA⁺ cations intercalate into the 2D material, we observe a gradual reduction in the PL intensity at the original emission wavelengths and the emergence of longer wavelength emission features. We interpret this to be due to FA⁺ exchanging with MA⁺ and BA⁺ and at first increasing the <*n*>-value in the now "triple cation" RPP perovskite, before the resultant film becomes predominantly FAPbI₃, or a mixed $FA_xMA_{1-x}PbI_3$ 3D perovskite, as judged by the emission peak at 780 nm. After spin-coating to remove excess conversion solution, subsequent thermal annealing at low (70 °C, 10 minutes) and then high (180 °C, 30 minutes) temperature completes conversion to the 3D perovskite, with the high temperature step selected with the intention of driving out remnant BA⁺ and MA⁺. The final emission peak of the annealed perovskite film is centred at 821 nm. Figure 2b shows the same process tracked by in-situ GIWAXS, highlighting the evolution of crystalline domains during conversion and subsequent annealing. After solution conversion, we find that only a small quantity of the 2D precursor phase is retained, with the majority having been converted directly to 3D perovskite. Reflections consistent with a very small quantity of the preferred RT phase (*2H*-FAPbI₃) are observed, however, these rapidly disappear along with remnant 2D intermediate upon high temperature curing. The direct conversion from 2D to 3D perovskite indicated by these combined in-situ GIWAXS and PL results indicate that the extended corner-sharing [PbI₆] octahedral network present in the precursor phase is retained into the 3D perovskite.

During in-situ GIWAXS measurement of the first five minutes of high temperature annealing, reflections corresponding to *6R*-PbI₂ gradually appear, but the intensity and position of these peaks stabilises – along with those of α-FAPbI₃ – consistent with complete volatilisation of small quantities of remnant MA⁺ from the 3D perovskite phase. As MA⁺ is present in the precursor phase, it is important to identify if any residual MA⁺ is left in the final annealed film, since if it were this could contribute to the ABX₃ phase stabilisation. Furthermore, previous work has indicated the phase stability of α-FAPbI₃ is

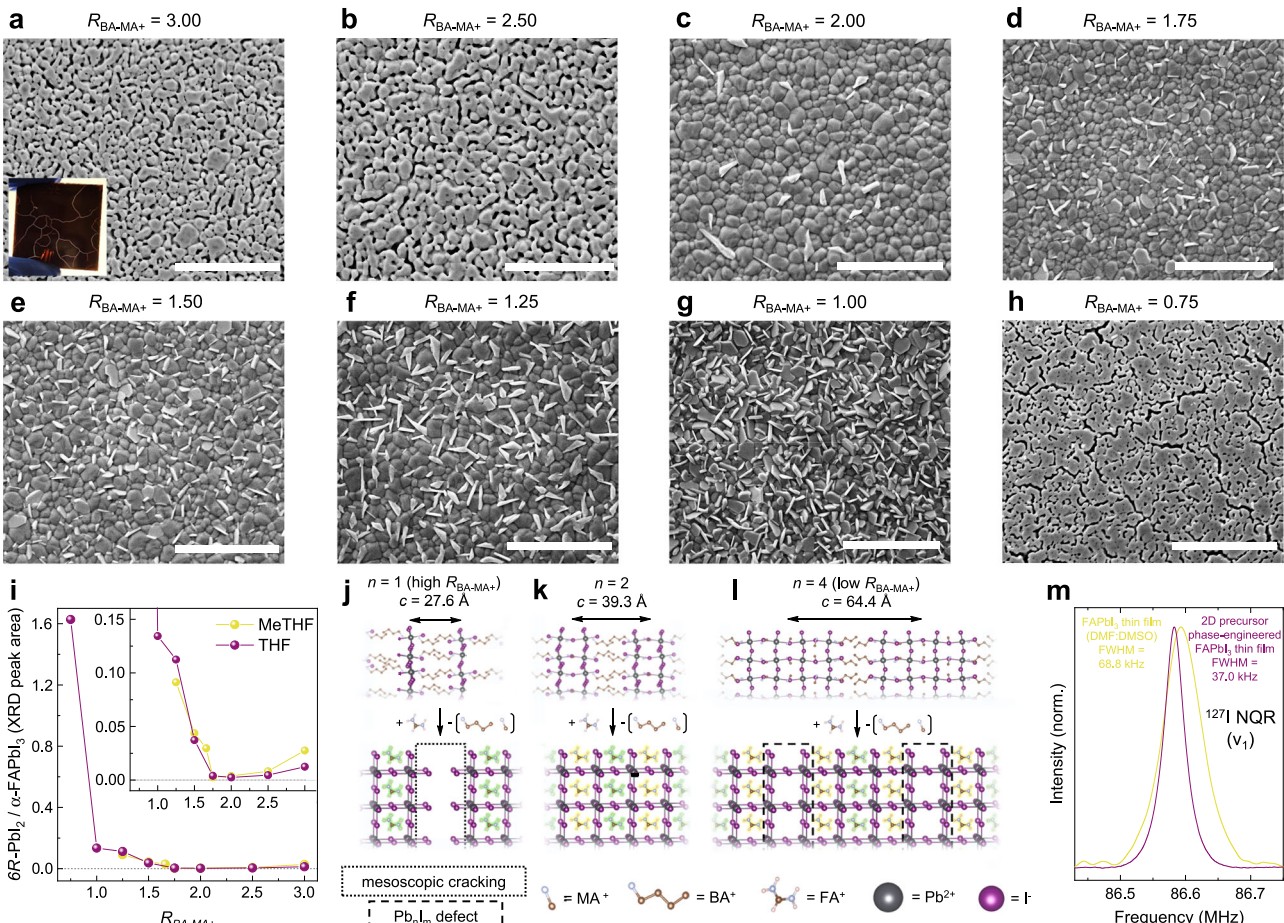

**Fig. 3 | Optimal 2D intermediate.** Scanning Electron Microscopy (SEM) images showing the microstructure of 2D-intermediate α-FAPbI$_3$ layers when a series of 2D intermediate processed from THF with varying $R_{BA\text{-}MA+}$ are converted sequentially. $R_{BA\text{-}MA+}$ = (**a**) 3.00 (inset: photograph showing macroscopic cracking of 3D perovskite layers converted from $R_{BA\text{-}MA+}$ = 3.00 intermediates due to extreme volume contraction), (**b**) 2.50, (**c**) 2.00, (**d**) 1.75, (**e**) 1.50, (**f**) 1.25, (**g**) 1.00, (**h**) 0.75. Scale bars: 5 μm. Schematic diagrams depicting the conversion of (**i**) BA$_2$PbI$_4$ ($n = 1$) (**j**) BA$_2$MAPb$_2$I$_7$ ($n = 2$), and (**k**) BA$_2$MA$_3$Pb$_4$I$_{13}$ ($n = 4$) into 3D α-FAPbI$_3$. FA$^+$ highlighted in green has replaced BA$^+$ in the organic channels of the 2D intermediate, while FA$^+$ highlighted in yellow has replaced MA$^+$ in the inorganic layers. Double-headed arrows display layer thickness. **l** Plot showing lead iodide (PbI$_2$) content as a phase fraction in comparison to 2D-intermediate α-FAPbI$_3$ extracted from XRD diffraction patterns (Supplementary Fig. 15) via peak integration of 6R-PbI$_2$ (001) and α-FAPbI$_3$ (100) scattering peaks. Inset shows an expanded view for low-PbI$_2$ region. **m** $^{127}$I nuclear quadrupole resonance (NQR) spectra of 2D-intermediate FAPbI$_3$ and FAPbI$_3$ (processed from DMF:DMSO) thin films, which have been mechanically exfoliated and powdered for measurement.

increased with the incorporation of some 2D phases[38], and hence determining if any residual BA$^+$ is present is also important. Ex-situ XRD characterisation of the fully-annealed 3D perovskite shows a reflection at 2θ = 13.92° (Supplementary Fig. 15a), consistent with that predicted for the (100) reflection of α-FAPbI$_3$ at 298 K (CCDC: 2243718; 2θ = 13.924°)[39]. We also perform $^1$H solution NMR spectroscopy measurements of the as-annealed 3D perovskite films dissolved in DMSO-$d_6$ (Fig. 2d, e), which confirm the absence of residual MA$^+$. The clear resolution of $^{13}$C satellite signals ($^1$J$_{1H\text{-}13C}$ coupling) indicates sensitivity to trace MA$^+$ or BA$^+$ as low as ~0.1 mol% (Supplementary Fig. 11), placing an upper limit residual MA$^+$ or BA$^+$ in our material. Further, in Supplementary Fig. 12 we show scanning electron diffraction (SED) measurements of the final annealed precursor phase-engineered 3D perovskite films. SED measurements allow detection and structural indexing of nanoscale crystalline domains that are too small for detection by bulk diffraction methods[40]. In these images we cannot find any diffraction spots which correspond to the unit cell dimensions expected from 2D perovskite phases. Thus, although this does not prove that there are no 2D domains present, we see no evidence that there are. From here on, we refer to our annealed 3D perovskite as α-FAPbI$_3$.

To further investigate the mechanism of 2D precursor phase-conversion we assess the impact of precursor composition on the resulting 3D perovskite. Scanning electron microscopy (SEM) images of α-FAPbI$_3$ processed via 2D precursor films of varying $<n>$ reveal a wide variety of microstructures. Use of a high $R_{BA\text{-}MA+}$ (BA:MA$^+$ precursor solution stoichiometry) 2D precursor phase (low $<n>$) results in a mesoporous structure in the 3D perovskite film (Fig. 3a-b), and even macroscopic cracking of the thin-films (Fig. 3a, inset). XRD analysis of α-FAPbI$_3$ processed from intermediate $R_{BA\text{-}MA+}$ ratios reveals how gradual reduction of $R_{BA\text{-}MA+}$ (toward higher $<n>$) leads to a pronounced increase in lead (II) iodide (6R-PbI$_2$) content in the as annealed films (Fig. 3i, Supplementary Fig. 14). We observe a consistent trend with the appearance of hexagonal 6R-PbI$_2$ platelets, which appear brighter in the SEM images (Fig. 3c–g)[41]. Conversion from high-$n$ 2D precursor phase films also leads to mesoscopic porosity visible via SEM (Fig. 3h). However, in contrast to low-$n$ precursor phases, XRD analysis of high-$n$-converted 3D materials demonstrates significant 6R-PbI$_2$ content (Fig. 3i). We rationalise these results by considering the careful balance of factors that control optimum $<n>$ for 2D precursor phase conversion and identify a 'Goldilocks region' in which optimal composition and morphology are achieved. We discuss our full

mechanistic rationalisation of the conversion process in detail in Supplementary Note 4. Here we summarise our findings.

For effective conversion of the precursor phase, we hypothesised that two processes should be optimised. (1) Facile intercalation of $FA^+$ cations into, and expulsion of $MA^+$ and $BA^+$ from the precursor phase must occur. (2) Volume contraction and structural rearrangement required upon conversion from the 2D perovskite to $\alpha$-$FAPbI_3$ must be minimised. Both these processes are, in turn, controlled by two properties of the 2D RPP precursor phase; spacer molecule channel density, and the thickness of the corner-sharing haloplumbate, $[PbI_6]_n$, layers in the 2D precursor phase. Both these properties are determined by the $<n>$ value.

The mesoscopic cracking observed in films converted from both high $<n>$ and low $<n>$ (low and high $R_{BA-MA+}$, respectively; Fig. 3a–h) is consistent with the release of local strain in the films during thermal curing. High $R_{BA-MA+}$ precursor phases possess high spacer channel density, which is expected to facilitate rapid cation exchange during conversion. However, 2D precursor films rich in the $n=1$ phase possess low lead halide density (density of $[PbI_6]$ octahedra), and thus require significant volume contraction upon conversion to form a continuous network of 3D corner-sharing $[PbI_6]$ octahedra. By this mechanism, volume contraction leads to local strain in the converted film, and thus mesoscopic cracking upon curing. We show this process schematically in Fig. 3j. Conversion of low $R_{BA-MA+}$ precursor phases to a 3D perovskite phase requires substantially less volume contraction. However, as shown in Fig. 3i, XRD analysis of converted $R_{BA-MA+} = 0.75$ films show that X-ray scattering intensity of the $6R$-$PbI_2$ is dominant over the 3D perovskite. XRD of the same series of 2D perovskites (Supplementary Fig. 2) revealed that the $R_{BA-MA+} = 0.75$ precursor phase is the only one to contain majority $n \geq 4$ RPP phases. The lead halide density of $PbI_2$ is even greater than of 3D perovskites and so conversion of a low $R_{BA-MA+}$ precursor phase to $PbI_2$ is also expected to result in substantial local strain, and thus mesoscopic cracking of the film, as observed in Fig. 3a.

Next, we consider the origin of $PbI_2$ in the final $FAPbI_3$ materials. For $n > 2$ precursor phases, complete extraction of $MA^+$ from, and intercalation of $FA^+$ cations into the intact haloplumbate structure requires cation migration across multiple perovskite A-sites, penetrating through unbroken layers of $[PbI_6]$ octahedra – which are not dissolved by the conversion solvent, $n$-butanol – and that act as an energetic barrier to ion intercalation. Any $MA^+$ not extracted from the precursor phase during conversion is expected to volatilise during subsequent thermal curing at 180 °C, via mechanisms discussed in Supplementary Note 8. As Fig. 3l illustrates, such a process yields $Pb^{2+}$ and $I^-$-rich regions in the forming 3D perovskite, and ultimately regions of crystalline $PbI_2$ upon grain ripening during thermal curing. The occurrence of such ripening is evidenced in the substantial evolution in morphology observed during high temperature curing (Supplementary Fig. 13g–j). This proposed mechanism is consistent with the gradual increase in $PbI_2$ content observed in converted $1.75 \geq R_{BA-MA+} \geq 0.75$ precursor phase films. To further demonstrate this effect, and the integral role spacer molecule channels play in conversion, we attempt the conversion of neat $MAPbI_3$ (representing $(BA^+)_2(MA^+)_{n-1}Pb_nI_{3n+1}$ with $n = \infty$; $R_{BA-MA+} = 0$) with the same solution conversion protocol. Absorbance spectra of this 'MAPbI$_3$ precursor phase' and the material resulting from its attempted conversion show the formation of $PbI_2$ in the converted layer, and a blue-shifted absorption onset compared to $\alpha$-$FAPbI_3$, confirming that complete conversion of the MAPbI$_3$ precursor phase cannot been achieved (Supplementary Fig. 16a). Cross-sectional SEM images of the MAPbI$_3$ precursor phase film and the converted film (Supplementary Fig. 16b, c) reveal the formation of significant voids at the bottom of the perovskite layer during conversion, similar to those observed in films converted from both high and low $<n>$ 2D precursor phases (Fig. 3a–h).

In order to estimate the strain induced or released when going from the precursor phases to the 3D perovskite, we first employ thin

film profilometry and simple structural considerations. Profilometer measurements indicate that the thickness of our optimum 2D precursor phase film ($R_{BA-MA+} = 1.5$; predominantly $n = 2$) contracts from 1,050 nm to 725 nm upon conversion to $\alpha$-$FAPbI_3$; a 32% volume contraction. Calculations using the reported unit cell volume of $BA_2MAPb_2I_7$ (3118.7 Å$^3$)[42], which contains eight $PbI_6^{4-}$ octahedra, and that of $FAPbI_3$ (256.4 Å$^3$)[43], indicate a theoretical volume contraction of approximately 34% should be observed. These data suggest that the majority of volume contraction upon conversion occurs in the vertical plane rather than laterally. This almost entirely accounts for the expected volume contraction due to the increased density of $PbI_6$ octahedra in the 3D perovskites. Thus, almost all the strain is released in this conversion process. To further confirm the absence of microstrain in our $\alpha$-$FAPbI_3$ thin films, we employ $^{127}I$ nuclear quadrupole resonance (NQR) spectroscopy[39,40]. As discussed in Supplementary Note 4, $^{127}I$ NQR is highly sensitive to local symmetry distortions in the $\alpha$-$FAPbI_3$ structure induced by residual strain, A-site mixing (e.g. incorporation of $MA^+$) or X-site mixing, which lead to a broadening of the NQR transition[39,40]. In comparison to $\alpha$-$FAPbI_3$ thin films made by a conventional DMF:DMSO solution processing route (in the absence of any additives), 2D precursor phase-engineered $\alpha$-$FAPbI_3$ thin films show a markedly narrower $^{127}I$ NQR transition (Fig. 3m). This is compelling evidence that none of the phenomena noted above occur in our material, notably the absence of residual strain and further confirms that our $\alpha$-$FAPbI_3$ is $MA^+$-free.

The mechanics of 2D precursor phase-engineered conversion and growth require significant further investigation. However, overall it is clear that, unlike intermediates featuring coordinating solvents or other lower-dimensionality materials reported to-date[34,35,44], the $<n>>$ 1 RPPs reported here hold two critical advantages as sequential deposition precursor phases for 3D perovskites: an extended corner-sharing haloplumbate structure already present in the precursor phase ready to serve as a scaffold for 3D perovskite growth, as evidenced by the direct conversion of 2D perovskite into $\alpha$, not $\delta$, phase $FAPbI_3$; and built-in organic channels within the lead-halide scaffold that can facilitate rapid intercalation of a conversion solution containing A-site cations.

We note that very recently Sidhik, et al. reported a related route to fabricate $\alpha$-phase $FAPbI_3$, using DMF:DMSO solvents, and also observed very high quality, $\alpha$-phase stable $FAPbI_3$[45]. Comparing our $\alpha$-$FAPbI_3$ to that published by Sidhik, et al., we determine a (cubic) lattice parameter of 6.341 Å by applying a Pawley fit (Supplementary Fig. S14), as compared to 6.369 Å reported by Sidhik, et al. The cubic lattice parameter of unstrained $\alpha$-$FAPbI_3$ single crystals has been reported as 6.355 Å[39]. The slightly larger lattice parameter of Sidhik, et al. may originate from the small fraction of retained $BA^+$ reported in their material. Here, we find no evidence of any $BA^+$ retained in our $\alpha$-$FAPbI_3$ material.

To investigate if our $\alpha$-$FAPbI_3$ films have useful optoelectronic properties, we compare them with $\alpha$-$FAPbI_3$ perovskites fabricated via two conventional solution processing approaches utilising DMF:DMSO solvent mixtures; with and without methylammonium chloride (MACl) additive[46]. MACl has been shown to improve perovskite material quality, stability and optoelectronic performance[46–48]. Time-resolved PL measurements reveal an order of magnitude improvement in PL lifetime for 2D precursor phase-engineered $\alpha$-$FAPbI_3$ compared with MACl-free $FAPbI_3$, and a factor of three enhancement compared to MACl-additive $FAPbI_3$ (Fig. 4a and Supplementary Fig. 19b). By fitting fluence-dependence of PL decays from films of the three materials (Supplementary Fig. 17), we determine a reduction in the monomolecular trap-assisted recombination rate constant, implying suppression of non-radiative recombination centres. Notably, for the conventional MACl-free $FAPbI_3$, much higher excitation fluences and charge-carrier densities are required for recombination dynamics to transition from a "trap-mediated" monomolecular decay regime to

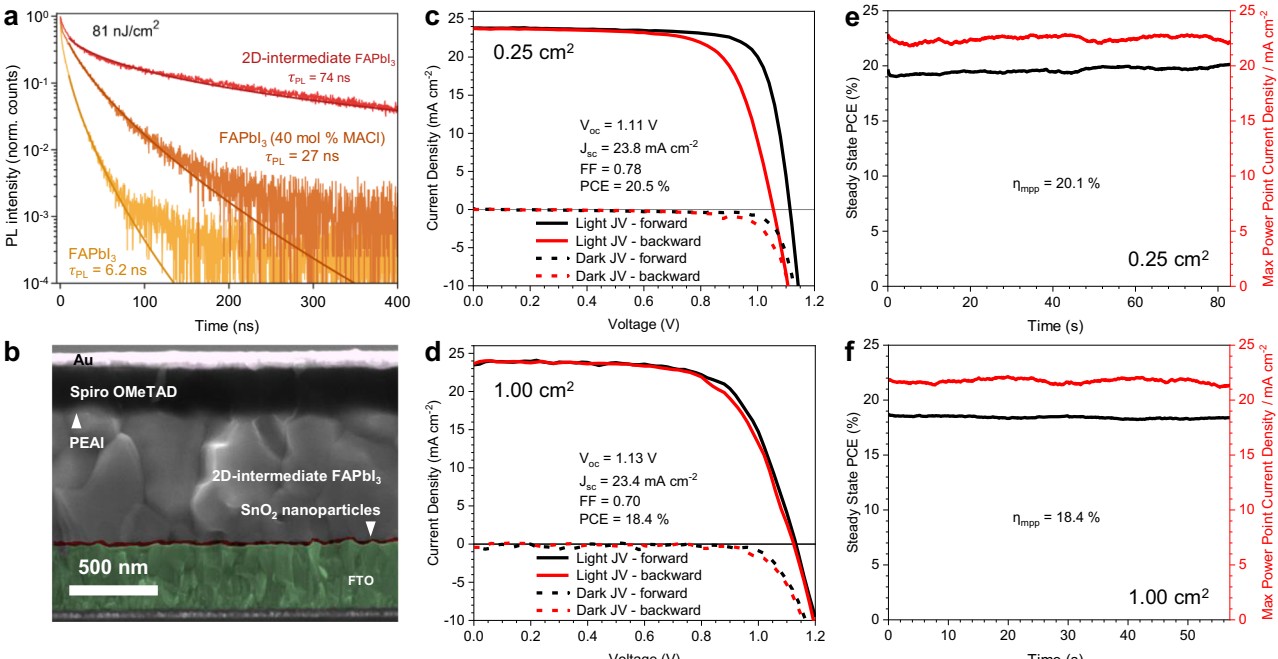

**Fig. 4 | PSC Operational Performance. a** Time-resolved photoluminescence (PL) transients of FAPbI$_3$ materials made using different fabrication methods. PL was detected at 810 nm following pulsed-laser excitations (398 nm at 2.5 MHz repetition rate) at an excitation fluence of 81 nJ cm$^{-2}$. PL lifetimes, $\tau_{PL}$, determined by stretched exponential fits are shown on the figure labelling. Solid lines are stretched-exponential fits to the measured data. **b** Cross-sectional scanning electron microscopy image of PSC fabricated utilising 2D precursor phase-engineered α-FAPbI$_3$. J-V characteristics and corresponding maximum power point tracked efficiency ($\eta_{mpp}$) and current density ($J_{mpp}$) measured at maximum power point for champion devices fabricated from 2D precursor phase-engineered α-FAPbI$_3$ with illuminated area 0.25 cm$^2$ (**c, e**) and 1.00 cm$^2$ (**d, f**). Light J-V and $\eta_{mpp}$ measurements were performed under simulated AM1.5 100.7 mWcm$^{-2}$ irradiance, accounting for the spectral mismatch factor. $J_{sc}$ = short-circuit current density, $V_{oc}$ = open-circuit voltage, FF = fill factor, $V_{mpp}$ = maximum power point voltage.

intrinsic bimolecular radiative recombination[49–51]. To probe the charge carrier mobility of these materials, we perform optical-pump-terahertz-probe (OPTP) photoconductivity spectroscopy (Supplementary Fig. 18) measurements. We determine that the effective sum of mobilities (electron and hole) and the bimolecular recombination rate $k_2$ of charge carriers, are largely unaffected by differences in film fabrication (Supplementary Fig. 19a). All the FAPbI$_3$ films studied have a very high mobility of approximately 60 cm$^2$V$^{-1}$ s$^{-1}$, which are some of the highest such values reported for thin film FAPbI$_3$ and attest to the generally high quality of the fabricated layers[52]. However, as a result of the highly effective suppression of trap-mediated charge-carrier recombination, for FAPbI$_3$ fabricated via the 2D precursor phase-engineered approach we calculate a significantly improved charge-carrier diffusion length under solar illumination conditions of 4.6 ± 0.2 μm, significantly higher than that for either FAPbI$_3$ material processed from DMF:DMSO (1.3 ± 0.1 μm and 3.0 ± 0.2 μm for neat and MACl additive-processed FAPbI$_3$) (Supplementary Fig. 19b).

To determine if the improved optoelectronic properties of the 2D precursor phase-engineered α-FAPbI$_3$ films translate into improved PSCs we fabricate n-i-p structured devices composed of FTO / SnO$_2$ / α-FAPbI$_3$ / PEAI passivation / spiro-OMeTAD / Au, which we show a cross-sectional SEM image of in Fig. 4b. In Fig. 4c-f we show the current-voltage curves and maximum power point tracked efficiency ($\eta_{mpp}$) for champion devices of both 0.25 cm$^2$ and 1 cm$^2$ respectively, and determine a $\eta_{mpp}$ of 20.1 % and 18.4 %, respectively. In Supplementary Fig. 20a we present the distribution of optimised 0.25 cm$^2$ and 1.00 cm$^2$ PSCs. For comparison, in Supplementary Fig. 20b-d we present PSCs fabricated similarly but employing FAPbI$_3$ with MACl additive (40 mol%), which achieve a maximum $\eta_{mpp}$ of 18.8 %.

### Investigating α-FAPbI$_3$ stability

Despite the favourable bandgap of α-FAPbI$_3$ for single-junction and for use as the middle junction in triple-junction PSCs, the well-known phase instability of this material is a drawback. At room temperature, the hexagonal non-perovskite *2H* (δ) phase of FAPbI$_3$ is thermodynamically preferred[53]. Although kinetic entrapment of the α-phase is possible, phase transformation is known to be accelerated by a range of external influences, including ambient humidity[4], and materials properties, such as iodide-defects[9]. In Supplementary Figs. 21–23 we show the evolution in absorbance, microstructure (by means of SEM and visible light microscopy) and phase composition (by XRD) for the same three FAPbI$_3$ materials investigated above, over 500 hours of storage under ambient conditions (18-22 °C, 20-55% relative humidity). We find that the composition of 2D precursor phase-engineered α-FAPbI$_3$ remains almost entirely unaltered over this period, while α-FAPbI$_3$ fabricated conventionally from DMF:DMSO solvents both with and without MACl additive show substantial secondary phase formation and reduction in optical density (absorbance), particularly at the band edge, over the first 300 hours of ambient storage.

In Supplementary Fig. 25 we show the unencapsulated "shelf stability" (stored in dry air, <10% relative humidity, at room temperature in darkness; adapted ISOS-D−1) of our n-i-p PSCs based on 2D precursor phase-engineered α-FAPbI$_3$, where we observe only minimal drop in performance over 18,000 hours (~ 2 years). Generally, such estimates of stability are of only limited usefulness in assessing long-term PSC stability[1]. However, in this instance, since α-FAPbI$_3$ is known to possess only kinetic metastability, such sustained performance confirms the unexpectedly promising phase-stability of our material.

To investigate the stability of our α-FAPbI$_3$ material under operational conditions and explore the relationship between perovskite processing route and resultant stability we perform in-situ XRD tracking the accelerated thermal degradation (130 °C, ~30% RH) of 2D precursor phase-engineered α-FAPbI$_3$, (Fig. 5a) which we discuss in detail in Supplementary Note 7. We find that α-FAPbI$_3$, MAPbI$_3$ and FA$_{0.83}$Cs$_{0.17}$Pb(I$_{0.9}$Br$_{0.1}$)$_3$ thin films fabricated via DMF:DMSO solvent systems all degrade to PbI$_2$ much more rapidly than our α-FAPbI$_3$

processed from highly volatile MeTHF, BA and BuOH solvents (Fig. 5b–d). Use of higher curing temperatures (180 °C) during processing reduces the rate of thermal degradation, but compromises perovskite crystallinity and phase purity (Fig. 5e-f). We also show that, besides 2D precursor phase-engineered $\alpha$-FAPbI$_3$, remarkably the most thermally stable perovskite composition investigated is MAPbI$_3$ processed from highly volatile ACN, which is much more thermally stable than MAPbI$_3$ via DMF:DMSO (Fig. 5g). From these results, we hypothesised that poor thermal stability may be due to retention of residual processing solvents in thin films processed from low volatility DMF and DMSO. To confirm that such residual processing solvents are indeed released during thermal degradation, we employ thermal desorption-gas chromatography-mass spectrometry (TD-GCMS), whereby volatile organic compounds (VOCs) are desorbed from perovskite thin films and individually characterised. These may be VOCs retained during processing or degradation products of the perovskite material. TD-GCMS is highly sensitive; we calculate the detection limit of DMSO retained within a 25 cm$^2$ thin film to be 0.002 wt.%[54] (Supplementary Note 7). We confirm that DMSO is released from $\alpha$-FAPbI$_3$, MAPbI$_3$ and FA$_{0.83}$Cs$_{0.17}$Pb(I$_{0.9}$Br$_{0.1}$)$_3$ films fabricated from DMF:DMSO. No solvents are released from either MAPbI$_3$ or our $\alpha$-FAPbI$_3$ processed from ACN and MeTHF, respectively, confirming a direct link between reduced perovskite thermal stability and release of residual processing solvents. Equally, we detect no low volatility residual solvents in $\alpha$-FAPbI$_3$ and FA$_{0.83}$Cs$_{0.17}$Pb(I$_{0.9}$Br$_{0.1}$)$_3$ processed via DMF:DMSO and cured at 180 °C. However, we do detect release of *sym*-triazine from these materials, as is the case in the degradation of every FA$^+$-containing perovskite investigated excluding 2D precursor phase-engineered $\alpha$-FAPbI$_3$ (15 different materials, Supplementary Fig. 28). A total of seven individual precursor phase-engineered $\alpha$-FAPbI$_3$ samples were analysed, with no *sym*-triazine detected. *Sym*-triazine has been reported as a thermal degradation product of FA$^+$[55–57]. By isolating a range of processing parameters, we identify that the 2D precursor phase-engineering crystallisation process itself is most likely responsible for the suppression of FA$^+$ volatilisation in our $\alpha$-FAPbI$_3$ (see discussion in Supplementary Note 7). We summarise these findings in Fig. 5h.

To compare the stability of our $\alpha$-FAPbI$_3$ to other highly stable perovskite compositions under conventional environmental stressing conditions we encapsulate perovskite films in glass-glass laminates employing an industry standard butyl-rubber edge seal, and subject the perovskite material to the conditions of two of the most demanding stability tests for perovskites; damp-heat (85 °C and 85 % relative humidity; ISOS-D-3) and light-soaking under elevated temperature (85 °C under a 1-sun equivalent light source; ISOS-L-2). Impressively, we observe very little change in the UV-vis absorption spectra for our $\alpha$-FAPbI$_3$ films when aged for over 3000 hours (3 times the IEC standard) under both stressing conditions. In Supplementary Fig. 32a, b, we show the optical density (OD) of the films, at wavelengths between 500 to 510 nm, as a function of stressing time. This gives a relatively sensitive indication of the evolution of "pin-holes" in the perovskite films, which have previously been observed to form during degradation[3,58]. The relatively high OD of our $\alpha$-FAPbI$_3$ films over the aging time, indicates that few pinholes are forming, which we confirm this with visible light microscopy, where in Fig. 5i–l we show images of the different films after >3000 hours stressing.

To assess if the improved material stability translates into improved device stability, we integrate our $\alpha$-FAPbI$_3$ into p-i-n structured PSCs. Although we have demonstrated >2 year "shelf stability" (Supplementary Fig. 25), n-i-p PSCs are known to suffer from a number of inherent instabilities under elevated moisture[59], temperature[60] and light[61], principally due to the instability of the transport materials usually employed in this architecture. However, reports of pristine FAPbI$_3$ in the more stable p-i-n configuration are unusual[62]. Furthermore, for 2D precursor phase-engineered $\alpha$-FAPbI$_3$ we find that

selection of the underlying hole transport material is severely limited by a combination of solvent incompatibility, high temperature processing of the perovskite and low mechanical adhesion with the as-deposited 2D precursor phase-engineered $\alpha$-FAPbI$_3$ layer. After mitigating these limitations (discussed in the Methods), we achieve substantial photovoltaic performance ($\eta_{mpp} = 18.8$ %, Supplementary Fig. 33).

Exposure of these devices to the ISOS-D-2 (85 °C, dark, N$_2$) aging procedure confirms the remarkable long-term thermal stability of our $\alpha$-FAPbI$_3$-based PSCs (Fig. 5a), retaining 95% of their initial $\eta_{mpp}$ after 1900 hours of aging (median of 15 cells, Supplementary Fig. 34). Next, we isolate the effect of light-induced degradation in the absence of elevated heat (adapted ISOS-L−1). Under these conditions we again find that our $\alpha$-FAPbI$_3$ PSCs are remarkably stable (projected $t_{80} = 3940$ hours, 21 cells, encapsulated, Supplementary Fig. 35). In comparison, PSCs of the same architecture employing FA$_{0.83}$Cs$_{0.17}$Pb(I$_{0.9}$Br$_{0.1}$)$_3$ show reduced photostability, with a $t_{80}$ of just 850 hours (10 cells). Although our target material is $\alpha$-FAPbI$_3$, we opt to compare this to an FA$_x$Cs$_{1-x}$ perovskite since these materials have been established as having state-of-the-art stability in single-junction PSCs[2,3]. Next, we subject similar PSCs to the ISOS-L-2 test (85 °C in ambient air, full spectrum 0.76 sun illumination with cells held at open circuit) in an Atlas Suntest CPS-Plus xenon-lamp aging box, as previously for thin film materials (Fig. 5i, j and Supplementary Fig. 32a, b). We find that device degradation of PSCs based on our precursor phase-engineered $\alpha$-FAPbI$_3$ achieve a promising ISOS-L-2 $t_{80}$ of 570 hours (8 cells) in comparison to FA$_{0.83}$Cs$_{0.17}$Pb(I$_{0.9}$Br$_{0.1}$)$_3$-based PSCs which exhibit a $t_{80}$ of just 40 hours (7 cells) under these conditions (Fig. 5m). We note that this is typical for a standard FA$_x$Cs$_{1-x}$ PSC aged under these conditions, without any stability-enhancing additives[2,3]. Our champion $\alpha$-FAPbI$_3$ cell (0.25 cm$^2$) achieves a $t_{80}$ of 800 hours. Considering the finding that our $\alpha$-FAPbI$_3$ thin films appear to be unchanged after 3600 hours of aging under ISOS-L-2 (Fig. 5i and Supplementary Fig. 32), we postulated that the 570-hour $t_{80}$ measured for PSCs based on this perovskite is still limited by factors besides photoabsorber bulk stability. Careful removal of the encapsulation and n-type contact layers after 1450 hours of ISOS-L-2 aging seems to confirm this inference. In Supplementary Fig. 37 we present XRD patterns of the aged devices, showing minimal change in the composition of either perovskite. However, SEM images of the pristine and aged perovskite layers (Supplementary Fig. 38) reveal degradation-induced void formation, which is substantially more severe in the FA$_{0.83}$Cs$_{0.17}$Pb(I$_{0.9}$Br$_{0.1}$)$_3$ PSCs.

Notably, these enhancements in PSC stability are achieved despite the presence of a small quantity of lead (II) iodide in the photoabsorber layer (Fig. 3i). *2H*-PbI$_2$ has been shown by us and others to generate defect-rich interfaces with $\alpha$-FAPbI$_3$ and to undergo photodegradation to metallic Pb$^0$ and I$_2$, producing iodide vacancies. As both iodide vacancies[63] and interstitial iodide[9] defects have been implicated in accelerating the rate of $\alpha$-to-$\delta$ phase transformation, the stability of our material is surprising. We note, however, that because of our high temperature processing, the PbI$_2$ phase present in our material is the *6R* polytype, which has recently been shown to form highly coherent, low-defect interfaces with $\alpha$-FAPbI$_3$[64]. Moreover, as Fig. 3a–h clearly show, the majority of *6R*-PbI$_2$ in 2D-intermediate $\alpha$-FAPbI$_3$ is in isolated grains. Thus, minimal PbI$_2$-FAPbI$_3$ interface is present in the material and any iodide vacancies generated by I$_2$ formation *in-operando* are isolated from the $\alpha$-FAPbI$_3$ and so can play no part in accelerating its phase transformation.

Finally, by developing a whole-cell encapsulation strategy (Supplementary Note 10) p-i-n PSCs employing 2D precursor phase-engineered $\alpha$-FAPbI$_3$ are able to pass the IEC61625-1 damp heat test (85 °C, 85% RH, ISOS-D-3), with >100% of initial median $\eta_{mpp}$ retained after 1930 hours ($t_{100}$ >1930 hours, 8 cells, Fig. 5n). By contrast, comparable FA$_{0.83}$Cs$_{0.17}$Pb(I$_{0.9}$Br$_{0.1}$)$_3$-based cells achieve a median $t_{95}$ of just 315 hours (8 cells). The IEC pass criterion is $t_{95}$ >1000 hours.

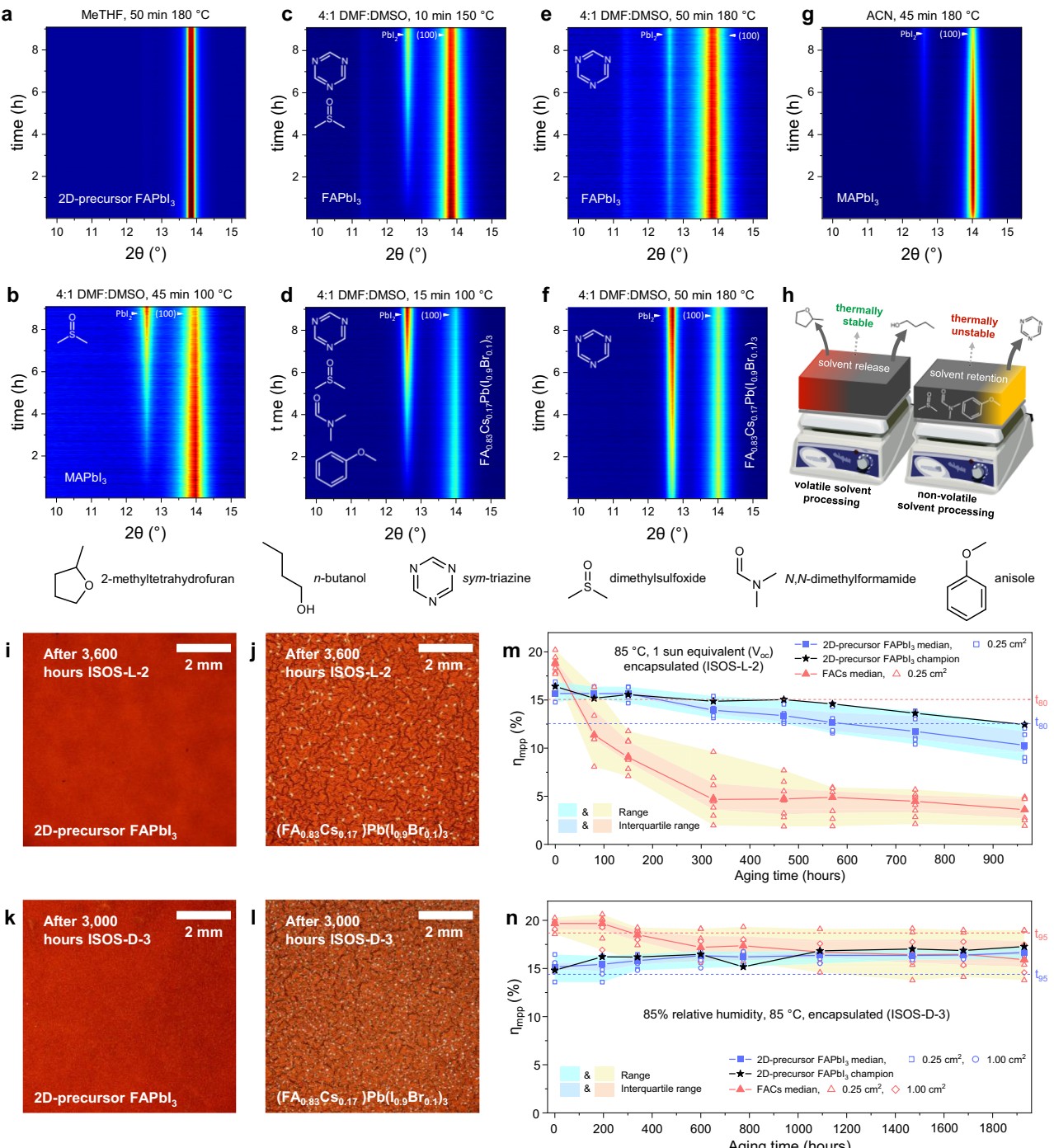

**Fig. 5 | Enhanced perovskite stability. a–g** Contour maps depicting in-situ x-ray diffraction (XRD) analysis of perovskite materials thermally degrading whilst held at 130 °C for 9 hours. Labels indicate perovskite composition and fabrication conditions (solvent, thermal curing). Full processing conditions described in Supplementary Note 6. Molecular structures shown correspond to the volatile species detected during thermal desorption-gas chromatography-mass spectrometry (TD-GCMS) analysis of each perovskite. **h** Schematic highlighting advantages to perovskite material properties of solution processing via MeTHF solvent system. Visible light microscopy images of encapsulated state-of-the-art perovskite thin films after aging for 3600 hours under 85 °C, 1-sun equivalent illumination (ISOS-L-2, **i-j**) and 3000 hours under 85 °C, 85 % relative humidity (ISOS-D-3, **k, l**) conditions.

Evolution under ISOS-L-2 (**m**) and ISOS-D-3 (**n**) stressing of maximum power tracked efficiency ($\eta_{mpp}$) measured periodically on p-i-n PSCs employing 2D precursor phase-engineered FAPbI$_3$ (7 cells and 8 cells, respectively) and FA$_{0.83}$Cs$_{0.17}$Pb(I$_{0.9}$Br$_{0.1}$)$_3$ (8 cells, under each condition, "FACs") as the photoactive layer. Cells undergoing ISOS-L-2 are encapsulated with a 250 nm layer of MoO$_3$ followed by an on-cell epoxy resin and cover slip. Cells undergoing ISOS-D-3 are encapsulated in glass-glass laminates employing an industry standard butyl-rubber edge sealant (described in Supplementary Note 10). Dashed lines show the median $t_{80}$ $\eta_{mpp}$ values for each dataset. The median, range and interquartile range of each dataset are shown, as well as each data point within them.

Notably, the survival of the 2D precursor phase-engineered α-FAPbI$_3$ for nearly 2000 hours under ISOS-D-3 conditions, despite the material's moisture-accelerated *2H*-phase instability, suggest that this simple three-ion perovskite may prove to be the most stable perovskite composition, matching the expectations for real world PV module deployment. To put our damp heat results into context, for copper indium gallium selenide PV modules, 3000 hours 85 °C, 85% RH stressing has been shown to be comparable to 20 years deployment in Miami, USA[65]. We summarise all lifetimes of PSCs presented in this work in Supplementary Table 3.

In conclusion, we have developed a volatile, low toxicity solvent system for α-FAPbI$_3$, one of the most desirable PV materials today, which enables fabrication of efficient and highly stable PSCs. This approach has allowed us to present an unconventional crystallisation pathway for a 3D perovskite in which the pre-existing corner-sharing [PbI$_6$]$^{4-}$ network present in a solid state 2D RPP perovskite is retained to direct the growth of α-FAPbI$_3$. Competitive PSC device efficiencies validate the 2D precursor phase-engineered crystallisation route, particularly considering the absence of MA$^+$ in the final 3D perovskite material and the use of a green solvent system. We find that α-FAPbI$_3$ fabricated by our processing approach possesses substantially improved thermal and phase-stability, as compared to a broad range of state-of-the-art perovskite compositions and processing routes. We demonstrate that this improved stability is the product of several factors. Critical among these is the 2D precursor phase crystallisation route itself, which both avoids the use of low volatility solvents – which we show are often retained in perovskite thin films, leading to long-term instability – and leads to a material in which FA$^+$ degradation to *sym*-triazine under thermal stressing is supressed. Integration of our 2D precursor phase-engineered α-FAPbI$_3$ into p-i-n PSCs confirm that the enhancement in bulk perovskite stability translates to impressive device stability. In particular, the photostability under elevated temperature (ISOS-L-2) of our α-FAPbI$_3$ gives a champion t$_{80}$ lifetime of 800 hours in comparison to just 100 hours for FA$_{0.83}$Cs$_{0.17}$Pb(I$_{0.9}$Br$_{0.1}$)$_3$-based PSCs. Further, we determine t$_{100}$ lifetimes of >1900 hours under damp heat stressing (ISOS-D-3). These damp heat results are particularly surprising since α-FAPbI$_3$ is considered to be metastable under ambient operational conditions. FAPbX$_3$ perovskites inherently bypass instability issues associated with A-site cation segregation and inhomogeneity[6], the possibility of creating Cs-rich impurity phases[7] and the hygroscopic nature of alkali-metal halide salts. Furthermore, by avoiding caesium, future material scarcity issues are avoided, representing a highly sustainable perovskite composition for terawatt scale PV deployment.

## Methods

### Materials
Fluorine-doped tin oxide-coated glass substrates (8 or 15 Ω cm$^{-2}$, AMG), tin (IV) oxide (15 wt. % in H$_2$O colloidal dispersion, Alfa Aesar), MeO-2PACz ([2-(3,6-Dimethoxy-9H-carbazol-9-yl)ethyl]phosphonic acid, >98.0%, Tokyo Chemical Industries), lead(II) iodide (99.99 %, trace metal basis, Tokyo Chemical Industries), methylammonium iodide (> 99.99 %, Greatcell Solar Materials), formamidinium iodide (> 99.99 %, Greatcell Solar Materials), phenethylammonium iodide (> 99 %, Greatcell Solar Materials), ethylenediammonium diiodide (Sigma Aldrich), PC$_{60}$BM ([6,6]-Phenyl-C61-butyric acid methyl ester), bathocuproine (BCP, >95.0%, Tokyo Chemical Industries), spiro-OMeTAD (2,2',7,7'-Tetrakis(N,N-di-p-methoxyphenylamino)-9,9'-spiro-obifluoren, >99.5 %, Luminescence Technology Corp.), FK209 Co(III) TFSI salt (tris(2-(1H-pyrazol-1-yl)-4-tert-butylpyridine)cobalt(III) tri[bis(trifluoromethane)sulfonimide], 98 %, Sigma Aldrich), bis(trifluoromethylsulfonyl)amine lithium salt (99.95 %, Sigma Aldrich), gold pellets (99.999 %, Kurt J. Lesker Company). Ethanol (200 proof, anhydrous, >99.5%, Sigma Aldrich), n-butylamine (99.5 %, Sigma Aldrich), 2-methyltetrahydrofuran (Biorenewable, anhydrous, >99.0 %, contains 250 ppm BHT as inhibitor, Sigma Aldrich), tetrahydrofuran (anhydrous, >99.9 %, contains 250 ppm BHT as inhibitor, Sigma Aldrich), n-butanol (anhydrous, 99.8 %, Sigma Aldrich), aluminium oxide nanoparticles (20 wt.% in 2-propanol, Sigma Aldrich), 2-propanol (anhydrous, 99.5 %, Sigma Aldrich), chlorobenzene (anhydrous, 99.8 %, Sigma Aldrich), acetonitrile (anhydrous, 99.8 %, Sigma Aldrich), 4-tert-butylpyridine (98 %, Sigma Aldrich), toluene (anhydrous, 99.8%, Sigma Aldrich), chlorobenzene (anhydrous, 99.8%, Sigma Aldrich), 1,2-dichlorobenzene (anhydrous, 99%, Sigma Aldrich). Prior to use, all FTO-coated substrates were scrubbed with an aqueous 2 vol % Decon 90 solution, rinsed with deionised water, sonicated in acetone, and subsequently sonicated in 2-propanol. Besides aluminium oxide nanoparticles, all other non-aqueous chemicals were stored in a N$_2$-filled glovebox before use and protected from exposure to light.

### n-i-p configuration perovskite solar cells
A colloidal suspension of tin oxide nanoparticles (400 μL) was diluted with ultrapure water (2600 μL). In an atmosphere containing minimum moisture (< 5 % relative humidity), 200 μL of this solution was placed statically on a UV-ozone treated (15 minutes) substrate coated with a fluorine-doped tin oxide layer (15 Ω cm$^{-2}$), then spun at 4000 rpm (1000 rpm s$^{-1}$) for 30 seconds before being immediately annealed at 150 °C for 30 minutes in the same environment. The substrates were allowed to cool, then immediately subjected to a further 15 minutes of UV-ozone treatment before being immediately used in the following processing step.

2-methyltetrahydrofuran (1500 μL) was added to lead iodide (1.266 mmol, 583.5 mg) and methylammonium iodide (1.139 mmol, 181.1 mg), followed by *n*-butylamine (1.899 mmol, 187.6 μL), and agitated until all solids were fully dissolved. This corresponds to an $R_{BA-MA^+}$ value of 1.67. 50 μL of this solution was spin-coated dynamically on top of the SnO$_2$ layer at 2500 rpm for 45 seconds in an environment at between 20–22 % relative humidity and <22 °C. The substrates were immediately annealed at 70 °C for 10 minutes in the same environment. After cooling, the substrates were coated with 350 μL of a 0.1 M solution of formamidinium iodide (0.500 mmol, 86.0 mg) dissolved in *n*-butanol (5000 μL). After 45 seconds of static soaking, the substrates were spun at 4000 rpm (1000 rpm s$^{-1}$) for 45 seconds, then immediately annealed for 10 minutes at 70 °C, followed by 30 minutes at 180 °C. We summarise this process schematically in Supplementary Fig. 1. Note: The full fabrication procedures for all other perovskite compositions utilised in this work are described in Supplementary Note 6.

The substrates were allowed to cool, then a 20 mM solution of phenethylammonium iodide (0.200 mmol, 49.8 mg) in 2-propanol (10,000 μL) was spin-coated dynamically on top at 5000 rpm for 45 seconds, in a N$_2$-containing glovebox.

A solution of spiro-OMeTAD (0.070 mmol, 85.8 mg) dissolved in chlorobenzene (1000 μL) and doped with 8.4 μL of a 0.250 M solution of tris(2-(1H-pyrazol-1-yl)-4-tert-butylpyridine)cobalt(III) tri[bis(trifluoromethane)sulfonimide] in acetonitrile, 19.4 μL of a 1.800 M solution of bis(trifluoromethylsulfonyl)amine lithium salt in acetonitrile, and *tert*-butyl pyridine (0.231 mmol, 38.0 μL). This solution was spin-coated on the PEAI-passivated substrates dynamically at 2500 rpm for 30 seconds, in a N$_2$-containing glovebox.

Finally, 75 nm of gold was evaporated on top of the substrates at an initial rate of 0.1 A s$^{-1}$ (to 5 nm, then ramped gradually to 1.0 A s$^{-1}$) at a pressure <2 × 10$^{-6}$ torr.

### p-i-n configuration perovskite solar cells
In a N$_2$ glovebox, a filtered (0.22 μm PTFE syringe filter) 0.33 mg mL$^{-1}$ solution of MeO-2PACz ([2-(3,6-Dimethoxy-9H-carbazol-9-yl)ethyl] phosphonic acid) in anhydrous ethanol (350 μL) was prepared and placed statically on a UV-ozone treated (30 minutes) substrate coated with a fluorine-doped tin oxide layer (15 Ω cm$^{-2}$), allowed to spread for 10 seconds, then spun at 3000 rpm (600 rpm s$^{-1}$) for 30 seconds

before being immediately annealed at 100 °C for 10 minutes in the same environment.

Separately, a colloidal suspension of alumina nanoparticles (50 μL) was diluted with 2-propanol (7500 μL) and ultrasonicated for at least 30 minutes ("1:150 $Al_2O_3$ NP:IPA"). This solution was filtered immediately before use (0.22 μm PTFE syringe filter). Preparation of this solution was carried out in ambient air, but all spin-coating was carried out in a $N_2$ glovebox.

The MeO-2PACz-coated substrates were allowed to cool, then in the same environment the solution of alumina nanoparticles (80 μL) was dispensed dynamically whilst the substrate was spun at 5000 rpm (5000 rpm s$^{-1}$) for 20 seconds. The substrate was then annealed at 100 °C for 5 minutes in the same environment. We have included this layer in the p-i-n device architecture in response to the poor wettability of perovskite solutions on carbazole-based SAMs, as has been reported elsewhere[66]. The use of $Al_2O_3$ nanoparticles in this way forms a thin mesoporous layer on top of the SAM, improving wettability[67].

2-methyltetrahydrofuran (1500 μL) and $n$-butylamine (1.688 mmol, 166.8 μL) were added to lead iodide (1.125 mmol, 518.6 mg) and methylammonium iodide (1.013 mmol, 161.0 mg) and agitated until all solids were fully dissolved. This corresponds to an $R_{BA-MA+}$ value of 1.67. 50 μL of this solution was spin-coated dynamically on top of the MeO-2PACz layer at 2500 rpm for 40 seconds in an environment at between 20–22 % relative humidity and <22 °C. The substrates were immediately annealed at 70 °C for 10 minutes in the same environment. After cooling, the substrates were coated with 350 μL of a 0.1 M solution of formamidinium iodide (0.500 mmol, 86.0 mg) dissolved in $n$-butanol (5000 μL). After 45 seconds of static soaking, the substrates were spun at 4000 rpm (1000 rpm s$^{-1}$) for 45 seconds, then immediately annealed for 10 minutes at 70 °C, followed by 30 minutes at 180–185 °C.

In ambient air, a colloidal suspension of alumina nanoparticles (50 μL) was diluted with 2-propanol (5000 μL) and ultrasonicated for at least 30 minutes ("1:100 $Al_2O_3$ NP:IPA")[67]. In a $N_2$ glovebox, a 0.5 mg mL$^{-1}$ solution of ethylenediammonium diiodide (EDAI$_2$) in a 1:1 mixture of anhydrous 2-propanol and toluene was prepared and heated for between 2 and 4 hours[68]. Also, a 30 mg mL$^{-1}$ solution of PC$_{60}$BM ([6,6]-Phenyl-C61-butyric acid methyl ester) in a 3:1 mixture of chlorobenzene:dichlorobenzene, and a 0.5 mg mL$^{-1}$ solution of bathocuproine (BCP) in anhydrous 2-propanol. All these solutions were filtered immediately before use (0.22 μm PTFE syringe filter).

In a $N_2$ glovebox, the solution of EDAI$_2$ (325 μL) was deposited on the on the previously deposited perovskite layer, and the substrate immediately spun at 3000 rpm (1333 rpm s$^{-1}$) to remove excess solution. The substrate was immediately annealed for 10 minutes at 100 °C in the same environment. EDAI$_2$ surface treatment has previously been reported as a highly effective strategy of improving both the $V_{oc}$ and operational stability of PSCs[69].

The substrate was allowed to cool, then the 1:150 $Al_2O_3$ NP:IPA solution (80 μL) was spin coated dynamically at 5000 rpm (5000 rpm s$^{-1}$), before being further annealed at 100 °C for 5 minutes in the same environment. Analogously to our use of $Al_2O_3$ nanoparticles to form a mesoporous underlayer beneath the perovskite, here we use the same strategy to combat the roughness of the perovskite surface. This roughness is associated with the large (>1500 nm) grains often observed in our FAPbI$_3$ material and leads to very poor adhesion and atomic-level contact between the PCBM electron transport layer and the perovskite. Recently in our group we have a developed a strategy by which a mesoporous alumina layer is introduced, and the PCBM solution allowed to intercalate into this structure prior to spin-coating (see below) to improve the contact between the perovskite and the fullerene layer[67]. We have also found that this layer is beneficial for long-term operational stability of the PSC, and as the aim of our p-i-n devices was to achieve the highest possible stability for control and target devices, this strategy was particularly selected for this reason[67].

The substrate was allowed to cool, then the PC$_{60}$BM solution (200 μL) was deposited on the substrate and allowed to spread for 10 seconds[67], before being spin coated at 2000 rpm (2000 rpm s$^{-1}$). The substrate was then annealed at 100 °C for 5 minutes in the same environment. Finally, once the substrates had cooled, the BCP solution (100 μL) was spin coated dynamically at 5000 rpm (5000 rpm s$^{-1}$), and the substrate annealed at 100 °C for 2 minutes in the same environment.

Using a chromium-coated tungsten bar, 3.5 nm of chromium was deposited on top of the substrate at a rate of 0.2 A s$^{-1}$, followed immediately by 100 nm of gold, which was evaporated on top of the substrates at an initial rate of 0.1 A s$^{-1}$ (to 5 nm, then ramped gradually to 1.2 A s$^{-1}$), all at a pressure <2 × 10$^{-6}$ torr. A layer of chromium was deposited prior to gold as this has previously been reported as reducing Au migration through to the perovskite layer thus increasing the long-term operational stability of the PV devices, which was our aim in fabricating these p-i-n configuration PSCs. We note, however, that use of this Cr layer typically reduces device PCE by -1%.

## Time-integrated photoluminescence

PL spectra were acquired using a time-correlated single photon counting (TCSPC) setup (FluoTime 300, PicoQuant GmbH). Samples were photoexcited using a 507 nm laser (LDH-P-C-510, Pico Quant GmbH) pulsed at a frequency of 40 MHz. The PL was dispersed using a monochromator, detected with hybrid photomultiplier detector assembly (PMA Hybrid 40, PicoQuant GmbH) and integrated over time.

## Ultraviolet-visible absorption spectroscopy

Reflectance and transmittance spectra were recorded on a Varian Cary 1050 UV Vis spectrophotometer equipped with an integrating sphere. From these measurements, in combination with the photoactive layer thickness, absorption coefficients were calculated assuming a direct bandgap semi-conductor. Separately, absorbance spectra were measured with a Varian Cary 300 Bio UV-visible spectrophotometer with a 50×50 mm reflective neutral density filter with an optical density of 3.0 (made of UV fused silica).

## Visible light microscopy

Optical microscope images were taken on a Nikon Eclipse LV100ND microscope with Nikon TU Plan Fluor lenses (10x/0.30 A, 20x/0.45 A, 50x/0.60 B, 100x/0.90 A). The images are taken with an attached Nikon Digital Camera D6.10.

## Time-correlated single photon counting (TCSPC)

Time-resolved PL of the thin films on quartz substrates was measured using TCSPC following excitation by a 398 nm picosecond pulsed diode laser at a repetition rate of 2.5 MHz (PicoHarp, LDH-D-C-405M). The resultant PL was collected and coupled into a grating spectrometer (Princeton Instruments, SP-2558), which directed the spectrally dispersed PL onto a photon-counting detector (PDM series from MPD), whose timing was controlled with a PicoHarp300 TCSPC event timer. The PL transients were measured at 810 nm, corresponding to the peak PL intensity in the steady-state PL spectrum of this composition.

The excitation fluence of 81 nJ cm$^{-2}$, corresponding to a charge-carrier density of the order of $10^{15}$ cm$^{-3}$, was used to ensure that the dominant recombination pathway is through monomolecular recombination. To account for both the time dependence and the local distribution of $k_1$ recombination rates, we fitted a stretched exponential function $I = I_0 e^{(\frac{t}{\tau_{char}})^\beta}$ to the PL intensity to account for the local distribution of monoexponential decay rates, whose effective lifetime is given by $\tau_{eff} = \frac{\tau_{char}}{\beta} \Gamma(\frac{1}{\beta})$. $\tau_{char}$ is the characteristic lifetime corresponding to the time taken for the PL intensity to drop to $I_0/e$, and $\beta$ is the distribution coefficient that encompasses the case for

monoexponential decays for the case $\beta = 1$, and the cases for a range of decay times (or potentially higher-order effects) as it decreases[70,71]. To account for the PL originating from electron-hole recombination we further double the effective fitted lifetime in order to extract the monomolecular charge-carrier recombination rate $k_1$, such that $k_1 = 1/2\tau_{eff}$, assuming $I \propto k_2 n^2$ at our excitation fluence of 81 nJ cm$^{-2}$ [50,51]. The samples were all mounted in a vacuum cell under low pressure ($\sim 10^{-2}$ mbar). We note that the absence of oxygen prevents the possible passivation of the photo-generated defects in the film and could have resulted in an underestimation of the lifetimes extractable under ambient operating conditions.

This spectroscopic technique initially generated charge-carriers closer to the surface of the film through which excitation occurs, owing to most of the 398 nm excitation being absorbed in the first 100 nm. This also accounts for the fast decay at very early times; most of the photogenerated charge-carriers will be initially generated closer to the surface where the density of trap states is higher than in the bulk, resulting in a rapid drop in PL intensity at very early times before the charge-carriers diffuse away further from the surface. The diffusion along the thickness of the film towards the bulk and perpendicular diffusion away from the excitation spot results in the local distribution of mono-exponential recombination rates assumed for our use of the stretch exponential fits.

### Optical-pump terahertz-probe (OPTP) spectroscopy

An amplified laser system (Spectra Physics, MaiTai – Ascend – Spitfire), with a 5 kHz repetition rate, centre wavelength of 800 nm and pulse duration of 35 fs is used to generate the THz radiation using a spintronic emitter. The THz probe is then focused onto the sample, overlaid with a 400 nm excitation pump that is generated using a Beta Barium Borate (BBO) crystal. The THz radiation transmitted through the sample is then detected via free-space electro-optical sampling in a ZnTe (110) crystal of thickness 200 μm[72].

### X-ray diffraction (XRD)

The 1D-XRD patterns were obtained with a Panalytical X'Pert Pro X-Ray diffractometer and In-situ 2D-XRD patterns (Fig. 5a–g, Supplementary Figs. 8 and 20) using a Rigaku SmartLab X-ray diffractometer and a HyPix-3000 2D hybrid pixel array detector, both with CuK$_{\alpha1}$ (1.54060 Å) source. A heating stage was employed in conjunction with the Rigaku SmartLab diffractometer for these measurements.

Grazing-incidence wide-angle X-ray scattering (GIWAXS) data (Supplementary Fig. 6) was acquired at the I07 beamline at Diamond Light Source. X-rays with energy 10 keV were incident on the samples at a grazing incidence angle $\alpha_i = 0.5$-$3°$, with scattering collected by a Pilatus 2 M (DECTRIS) hybrid photon-counting detector with a sample-to-detector distance (SDD) of 375 mm, with the geometry calibrated using LaB$_6$. In-situ GIWAXS data (Fig. 2b) was measured using the same configuration, with spin-coated 2D intermediate samples aligned before FAI solutions were deposited on top using an in-situ blade coater in ambient conditions. The coating apparatus incorporates a syringe driver, motorised blade, integrated hotplate and an N2 outlet directed at the sample to remove excess solvent after conversion. Solutions were deposited via a microfluidic tube adjacent to the measured area, and coated perpendicular to the beam with the blade set with a shim height of 500 μm and coating speed of -10 mm s$^{-1}$, and measured with a grazing incidence angle of 2°.

Additional GIWAXS data (Supplementary Fig. 24) was acquired with a SmartLab (RIGAKU) diffractometer with a 3 kW Cu X-ray source (8.04 keV) in parallel beam configuration with 0.5° pinhole optics, a 1° in-plane parallel slit collimator and 0.3° long collimator attachments. Scattering was collected with a HyPix-3000 (RIGAKU) 2D detector with 65 mm SDD, also at $\alpha_i = 2°$. For each measurement, the detector goniometer arm was rotated through 2θ angles from 0° to 40° in 1°

steps, with 15 min acquisition at each position, with detector images then remapped into Q-space and combined together. All data reduction was performed using scripts based on the PyFAI and pygix libraries[73,74].

### Scanning electron microscopy (SEM)

A FEI Quanta 600 FEG Environmental Scanning Electron Microscope (ESEM) was employed to investigate perovskite layer morphology. Accelerating voltages between 4-15 kV were employed for various analyses.

### Nuclear magnetic resonance (NMR) spectroscopy

A two-channel Bruker Avance III HD Nanobay 400 MHz instrument running TOPSPIN 3 equipped with a 5 mm z-gradient broadband/fluorine observation probe is used. The signal from residual non-deuterated DMSO solvent is used for reference.

### Nuclear quadrupole resonance (NQR) spectroscopy

For $^{127}$I NQR spectroscopy, thin films were mechanically exfoliated from FTO (2D-intermediate FAPbI$_3$) or glass (FAPbI$_3$, DMF:DMSO) substrates using a razor blade to produce powders. These thin film powders were packed into 2.5 mm zirconia rotors and compacted. $^{127}$I NQR spectra were recorded on a Bruker Avance III spectrometer equipped with a 2.5 mm CPMAS probe using 42 kHz RF field amplitude, a recycle delay of 0.05 s and with the probe placed outside of any external magnetic field. The sample rotor was static during the measurement.

### Thermal desorption-gas chromatography-mass spectrometry (TD-GCMS)

Thin films (<1 μm) of perovskite material on fluorine-doped tin oxide-coated glass substrates were loaded into a thermal extractor unit (Micro-Chamber/Thermal Extractor M-CTE250, Markes International)and heated at 165 °C for 60 minutes under a flow of N$_2$ gas (50 mL min$^{-1}$). Volatile components released during extraction were collected onto sorbent-packed collection tubes. The sorbent tubes are then loaded into a thermal desorption unit and heated rapidly to desorb the volatiles concentrated in the tube, which are then passed via N$_2$ carrier gas into the gas chromatography-mass spectrometry instrument (Agilent 5977B GC/MSD). Identification of the volatile components was done by comparison to the NIST 17 Mass Spectral Library. All reported species showed a > 85 % match with the database compound.

### Optically-modulated surface photovoltage measurements (SPV)

Modulated SPV spectra were measured in the configuration of a parallel plate capacitor (quartz cylinder partially coated with the SnO$_2$:F electrode, mica sheet as insulator), under ambient atmosphere. The SPV signal is defined as the change in the surface potential as a result of the illumination. In our case the Illumination was provided by a Halogen lamp, coupled to a quartz prism monochromator (SPM2), and modulated at a frequency of 8 Hz by using an optical chopper. The overall SPV amplitude (combined of the in-phase and out of phase SPV signals) were detected with a high-impedance buffer and a double phase lock-in amplifier (EG&G 5210).

### Characterisation of solar cells

Current-voltage (J–V) and maximum power point (MPP) measurements were measured (2400 series source meter, Keithley Instruments) in ambient air under both light (simulated AM 1.5 irradiance generated by a Wavelabs SINUS-220 simulator) and in the dark. The active area of the solar cell was masked with a black-anodised metal aperture to either 0.25 or 1.00 cm$^2$, within a light-tight holder. The 'forward' J–V scans were measured from forward bias to short-circuit and the 'backward' scans were from short-circuit to forward bias, both

at a scan rate of 245 mV s$^{-1}$. Active MPP tracking measurements using a gradient descent algorithm were performed for at least 30 s to obtain the steady-state power conversion efficiency. Some cells exhibited improvement over multiple measurements, in which case the peak performance was reported. This typically took two to five J–V scan plus MPP tracking iterations for the highest efficiency cells, in a measurement time of around 2–5 minutes. The intensity of the solar simulator was set periodically such that the short-circuit current density from a KG3-filtered Si reference photodiode (Fraunhofer ISE) matched its 1-sun certified value. A local measurement of the intensity before each batch of solar cell measurements were performed, was made by integrating the spectrum obtained from the solar simulator's internal spectrometer. By taking the ratio of this internal intensity measurement to one obtained at the time of calibration we determined the equivalent irradiance at the time of measurement. For the data presented in this publication, this gave values ranging from 0.985-1.005 suns equivalent, which have been applied to the calculation of power conversion efficiencies for each individual measurement. The spectral mismatch factor was estimated to be 1.022 according to a previously reported method[75]. This has also been applied to calculate power conversion efficiencies. We estimate the systematic error of this setup to be on the order of ± 5% (relative).

### External quantum efficiency (EQE)

External quantum efficiency (EQE) measurements were performed using a custom setup. The solar cells were illuminated with a 250 W quartz-tungsten halogen lamp that was first passed through a monochromator (Princeton Instruments SP2150) with a filter wheel (Princeton Instruments FA2448), then chopped with an optical chopper (Thorlabs MC2000B) at 280 Hz, and finally focussed onto the sample with a smaller spot size than the solar cell area (as defined by the metallic top contact). The amplitude of the resulting AC current signal was measured with a lock-in amplifier (Stanford Research Systems SR830) as the voltage drop across a 50 Ohm resistor in series with the solar cell. A reference measurement of a calibrated Si photodiode of known EQE (Thorlabs FDS100-CAL) was used to correct the measured signal from the solar cell to absolute units. White light bias was applied using LEDs.

### Operational stability accelerated-aging equipment

PSCs (p-i-n configuration) aged under ISOS-L-1 and ISOS-L-2 conditions were encapsulated using a cover glass and UV-curable epoxy adhesive (Everlight Eversolar AB341), which was spread across the full active area of the PSCs. As discussed in Supplementary Note 9, for ISOS-L-2 testing, in addition to the PSC fabrication described above, prior to on-cell encapsulation (where the encapsulation glue is spread across the full device area) a 250 nm layer of $MoO_3$ was thermally evaporated[76]. For ISOS-D-3 aging, an industry standard edge-sealant was employed to encapsulate PSCs, as described in Supplementary Note 10.

For ISOS-D-2 aging, PSCs were not encapsulated and were placed in an oven controlled at 85 °C in a $N_2$ filled glovebox.

ISOS-D-3 aging was carried out in a Weiss Technik LabEvent L C/64/40/3 environmental testing chamber at 85 °C and 85% relative humidity (RH).

For adapted ISOS-L-1 aging, a bespoke setup was used. RS Peltier Pro modules were employed to actively cool the PSCs to approximately 22 °C (measured by a photodiode encased in a black thermally conductive material to simulate the conditions in the photoactive layer). As shown in Supplementary Fig. 29b, an LED array positioned such as to provide 0.86 sun equivalents to the surface of the PSCs was employed. Aging was carried out at $V_{oc}$ under these conditions.

For ISOS-L-2 aging an Atlas SUNTEST CPS+ light-soaking chamber under simulated full spectrum AM1.5 sunlight was employed (no UV filter was applied during the aging process). Aging was carried out at $V_{oc}$ under these conditions. The temperature for the aging chamber was measured by a black standard temperature control unit.

To perform J-V characterisation and stabilised $V_{oc}$, $J_{sc}$ and active MPP tracking measurements, all substrates were removed from the aging setups and immediately tested before being returned to the relevant aging condition.

### Reporting summary

Further information on research design is available in the Nature Portfolio Reporting Summary linked to this article.

## Data availability

All relevant data are provided in the figures, tables and Supplementary Information. The raw data that support the findings of this study are available via the Zenodo repository at: https://doi.org/10.5281/zenodo.14008401.

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

## Acknowledgements

We acknowledge the contribution of Shuan Reeksting at the MC[2] Institute, University of Bath, for performing and advising on our TD-GCMS measurements, and Kilian Lohmann for providing the evaporated $FAPbI_3$ sample for TD-GCMS. Further, we acknowledge Diamond Light Source for time on beamline I07 under proposal SI33462-1 for the GIWAXS measurements and thank Jonathan Rawle for assistance with these measurements. This project was supported by the Plastic Electronics Centre for Doctoral Training, funded by EPSRC UK, the EPSRC grant number EP/V027131/1 (B.M.G.) and has received funding from the European Union's Horizon 2020 research and innovation programme under the Marie Skłodowska-Curie grant agreement No 764787 (P.H.). D.J.K. acknowledges the ERC-selected UKRI Horizon Europe guarantee funding (PhotoPeroNMR - grant agreement number EP/Y01376X/1). I.L. thanks the AiF project (ZIM-KK5085302DF0) for financial support.

## Author contributions

B.M.G. conceived the experiments with support from P.H. and J.A.S. B.M.G. developed the ether-amine solvent systems, carried out characterisation of the 2D perovskites, investigation of the mechanism of 2D-intermediate conversion, fabrication of the perovskite solar cells, TD-GCMS analysis and all materials and device stability measurements besides those involving in-situ XRD measurements. B.M.G. also carried out the NQR measurements, supervised by D.J.K. S.C. and P.C. assisted with fabrication of p-i-n configuration solar cells. B.M.G., P.H. and J.A.S. carried out XRD analysis. K.A.E. carried out all OPTP and TCSPC measurements and associated calculations, supervised by L.M.H. J.A.S., B.M.G. and F.Y. conducted the GIWAXS characterisation and analysis. D.T.W.T, R.C.K. and A.K.A.Z. designed and built the GIWAXS setup. A.S. analysed the SED data, supervised by N.K.N. E.Y-H.H carried out Pawley fitting of XRD of optimised material. J.M.B. constructed the setup used for EQE measurements and developed the software used to measure and analyse all PSC device performance. M.G.C. designed and implemented the solar simulator and cell layout used for all device performance measurements. I.L. carried out and analysed the modulated SPV measurements. H.J.S supervised the project, developed the ideas and concepts with B.M.G, and raised the funds for the work. B.M.G. wrote the manuscript and P.H., J.A.S. and H.J.S. edited it. All authors contributed to proofing of the manuscript and writing/editing sections related to their contributions.

## Competing interests

H.J.S. is a co-founder and Chief Scientific Officer of Oxford PV Ltd., a company industrialising perovskite PV. The other authors declare no competing interests.
