## [Peer Review file · Nature Communications]

A green solvent enables precursor phase engineering of stable formamidinium lead triiodide perovskite solar cells

Corresponding Author: Professor Henry Snaith

This manuscript has been previously reviewed at another journal. This document only contains reviewer comments, rebuttal and decision letters for versions considered at Nature Communications.

Version 0:

Reviewer comments:

Reviewer #1

(Remarks to the Author)

The manuscript by Gallant and co-workers shows the stabilization of FAPI using 2D-perovskite precursors using a two-step processed. The quasi 2D film appears to have converted to a FAPI film. From the referee exchanges previously, it appears that the authors have addressed almost all concerns. From the exchanges, it also appears that this paper has been through several rounds of review. Without inducing delays in publishing this, below are some comments that I feel the authors must consider given that new work has appeared.

1. Given the recent work DOI: 10.1126/science.abq699 , which shows a clear 1.48 eV band edge in the absorbance and ~1.49 eV step change in the EQE, which is still lower or equal to 1.5 eV compared to this work with a band-gap of 1.53 eV. What do the authors think causes this discrepancy if the films are indeed pure? Could it be that the films are quasi-2D with with undetectable amount of BA? Or could it be that the degree of distortions is larger, which then increases the bandgap ?
2. What is the lattice constant of the final film? I understand that the authors have retracted their claim of templating, but it might be instructive to compare this with the above work.

In the final analysis, this is exciting work and it clearly shows that 2D perovskites have a critical role to play in improving the stability of 3D perovskites and achieving "pure" phases. recommend publishing this work once these comments have been addressed.

Reviewer #5

(Remarks to the Author)

The manuscript is originally o Nature Energy, then the rebuttal version is to Nature Communications. I read the revised manuscript and rebuttal letter carefully and think it is of high quality for Nature Comm. after addressing the reviewers' comments. The added experimental results and explanations are largely sound. The proposed biorenewable solvent system is interesting to the field, and the 2-step process of forming 2D precursor film, them convert to 3D FAPbI₃ film is much better explained after the revision. While the efficiency is not very high, with the mechanism explained with evidence, and good stability data, I think it is to the standard of Nature Comm. I suggest the acceptance of the manuscript once the paper presentation is carefully checked. For example, in the conclusion section, the first sentence missed ".". The term "green solvent" is suggested to change to "biorenewable solvent" to make it scientifically solid.

A green solvent system for precursor phase-engineered sequential deposition of stable formamidinium lead triiodide for perovskite solar cells

Reviewer Response

Reviewer #1 (Remarks to the Author)

The manuscript by Gallant and co-workers shows the stabilization of FAPI using 2D-perovskite precursors using a two-step processed. The quasi 2D film appears to have converted to a FAPI film. From the referee exchanges previously, it appears that the authors have addressed almost all concerns. From the exchanges, it also appears that this paper has been through several rounds of review. Without inducing delays in publishing this, below are some comments that I feel the authors must consider given that new work has appeared.

1. Given the recent work DOI: [10.1126/science.abq699](https://doi.org/10.1126/science.abq699), which shows a clear 1.48 eV band edge in the absorbance and ~1.49 eV step change in the EQE, which is still lower or equal to 1.5 eV compared to this work with a band-gap of 1.53 eV. What do the authors think causes this discrepancy if the films are indeed pure? Could it be that the films are quasi-2D with with undetectable amount of BA? Or could it be that the degree of distortions is larger, which then increases the bandgap ?
- We are grateful to the Reviewer for their positive assessment of our work, and for the consideration of this work particularly in the context of the recent study from Mohite and co-workers. Although the Reviewer did not highlight the similarities explicitly, we would like to take the opportunity to highlight that our work contains substantial novelty over that of Mohite and co-workers, in particular the development of our green solvent system, a fully sequential processing route, the extensive characterisation of the precursor phase-engineered growth mechanism (added in previous rounds of review), and the in-depth study of the origin of stability enhancement imbued into our thin film FAPbI₃ material by this growth process. Furthermore, we show proper statistics from our stability measurements, rather than data for a single solar cell.
 - As we believe the Reviewer has seen, we have discussed in substantial detail the variability in the reports of perovskite semiconductor bandgaps, in particular those for FAPbI₃. In brief, this variability comes as a result of experimental discrepancies in bandgap calculations as applied by different authors and variability in the perovskite material (thin film thickness, single crystals, etc.). The most relevant band gap for PV cells is the so called “PV band gap” which can be derived by taking the differential of the EQE spectrum and noting the peak of this differential. This corresponds to the steepest part of the EQE spectrum. This PV gap most closely represents a hypothetical step-function band gap, which is often used for calculating detailed balance voltage and efficiency limits. I direct the reviewer to Nayak et al. Nature Review Materials 2018 for a full discussion on this topic. Unfortunately, despite this being pretty well recognised by the field now, Mohite et al. used the Tauc analysis to define their band gap, which usually gives a slightly different band gap than the PV gap, where the latter depends upon the thickness of the absorber layer and the optical properties of the entire cell. The onset of the EQE is also not the appropriate energy to identify as the band gap for a PV cell. Moreover, it appears that Mohite et al. did not take into account thin film reflectance when calculating the spectrally-resolved absorption coefficient, $\alpha(\lambda)$, of their thin films. This can lead to non-zero sub-bandgap absorption coefficients, which in turn result in an underestimation of the semiconductor bandgap. Noting further, for Tauc analysis $(\alpha \cdot h \cdot f)^2$ (h = Planck's constant, f = frequency) should be plotted on a linear scale against the energy, in eV. A linear region near the zero line should then be fitted, and the Tauc bandgap defined as the intersect of this line with the x-axis. For reasons that are not clear, probably due to excessive light scattering, Mohite and co-workers have decided to plot their graphs on a log-linear scale, and taken the intercept of a linear line of the higher energy and lower energy sections of the graph to identify the Tauc gap. This is highly unusual so we can not comment on the accuracy of their reported “band gap”, but it does not appear to be a Tauc bandgap.

- Falling short of extracting Mohite et al.'s data and analysing it, we have copied their reported EQE spectrum below, and inserted a dashed green arrow at approximately the steepest part of the EQE spectrum, which intercepts the x-axis somewhere between 810 and 820nm. This corresponds to a PV band gap of 1.53 to 1.51 eV.

As we have previously responded to the question concerning of evidence for our sample being neat FAPbI₃, we won't repeat it here, other than to simply reaffirm that we are as certain as we can be that our material is neat FAPbI₃.

2. What is the lattice constant of the final film? I understand that the authors have retracted their claim of templating, but it might be instructive to compare this with the above work.

- The Reviewer raises a very interesting point, however, we haven't carried out a full Reitfeld/Pauli fitting of the PXRD patterns reported here (partially as they show very strong preferred orientation), we did not report a lattice spacing, instead we have more simply reported the 2θ value of the diagnostic (100) planes.
- In response to the Reviewer's comment, we have fitted the 1D XRD peaks corresponding to our FAPbI₃ material individually and report the 2θ values with corresponding implied d spacing. We emphasise that we do not present this as a truly fitting of the diffraction pattern.

Reflection (hkl)	Experimental 2θ (°)	d spacing (Å)	Lattice parameter (Å)
(100)	13.916	6.3587	6.3587
(110)	19.739	4.4940	6.3555
(111)	24.246	3.6679	6.3530
(200)	28.06	3.1774	6.3548
(210)	31.465	2.8409	6.3524
(211)	34.551	2.5939	6.3537
Mean	-	-	6.3547 ± 0.0021

- This gives an implied cubic lattice parameter of 6.355 Å, on average, consistent with CCDC: 2243718 for pure FAPbI₃ single crystals; $a = 6.355$ Å. This is somewhat less than that reported by Mohite and co-workers (6.369 Å), perhaps due to the presence of BA⁺ remnant in their material, but is within the 'strain-free α -FAPbI₃' region that these authors report and thus consistent with our NQR measurements (**Figure 3m**) that confirm the absence of residual strain in our FAPbI₃ thin films. We have now discussed the difference between the results of Mohite and co-workers in the main text, and added the above analysis to the Supplementary Information (**Supplementary Figure S14**).

In the final analysis, this is exciting work and it clearly shows that 2D perovskites have a critical role to play in improving the stability of 3D perovskites and achieving "pure" phases. recommend publishing this work once these comments have been addressed.

We thank the Reviewer for considering our manuscript favourably.

Reviewer #5 (Remarks to the Author):

The manuscript is originally o Nature Energy, then the rebuttal version is to Nature Communications. I read the revised manuscript and rebuttal letter carefully and think it is of high quality for Nature Comm. after addressing the reviewers' comments. The added experimental results and explanations are largely sound. The proposed biorenewable solvent system is interesting to the field, and the 2-step process of forming 2D precursor film, them convert to 3D FAPbI₃ film is much better explained after the revision. While the efficiency is not very high, with the mechanism explained with evidence, and good stability data, I think it is to the standard of Nature Comm. I suggest the acceptance of the manuscript once the paper presentation is carefully checked. For example, in the conclusion section, the first sentence missed ".". The term "green solvent" is suggested to change to "biorenewable solvent" to make it scientifically solid.

We thank the Reviewer for their positive appraisal of our work, and for the formatting corrections. We have checked the manuscript and confirmed that it correct and ready for presentation.

We note that we have used the term "green solvent", which is in common use in our field (e.g. s41560-022-01086-7), to summarise two important aspects of the solvent system presented. Namely, the biorenewable production of MeTHF and the greatly reduced toxicity of the overall solvent system (9:1 MeTHF:BA) in comparison to established perovskite solvent systems (**Figure 1a**).